# Engineered Cationic Antimicrobial Peptides (eCAPs) to Combat Multidrug-Resistant Bacteria

**DOI:** 10.3390/pharmaceutics12060501

**Published:** 2020-05-30

**Authors:** Berthony Deslouches, Ronald C. Montelaro, Ken L. Urish, Yuanpu P. Di

**Affiliations:** 1Department of Environmental and Occupational Health, University of Pittsburgh Graduate School of Public Health, Pittsburgh, PA 15261, USA; peterdi@pitt.edu; 2Department of Microbiology and Molecular Genetics, University of Pittsburgh School of Medicine, Pittsburgh, PA 15219, USA; rmont@pitt.edu; 3Department of Orthopaedic Surgery, University of Pittsburgh, Pittsburgh, PA 15213, USA; urishk2@upmc.edu

**Keywords:** antimicrobial peptides, cationic amphipathic peptides, host defense peptides, gene-encoded antimicrobial peptides, helical antimicrobial peptides, polymyxins, colistin, synergy, antibiotic resistance, multidrug-resistant bacteria, ESKAPE pathogens, antibiotics, peptide antibiotics

## Abstract

The increasing rate of antibiotic resistance constitutes a global health crisis. Antimicrobial peptides (AMPs) have the property to selectively kill bacteria regardless of resistance to traditional antibiotics. However, several challenges (e.g., reduced activity in the presence of serum and lack of efficacy in vivo) to clinical development need to be overcome. In the last two decades, we have addressed many of those challenges by engineering cationic AMPs *de novo* for optimization under test conditions that typically inhibit the activities of natural AMPs, including systemic efficacy. We reviewed some of the most promising data of the last two decades in the context of the advancement of the field of helical AMPs toward clinical development.

## 1. Introduction

The development and the clinical use of the traditional classes of antibiotics, both prophylactically and therapeutically, have facilitated many of the advances in modern medicine. However, over the past three decades, the alarming rate of multidrug-resistance (MDR) has threatened to end the antibiotic era [1,2]. If this trend continues without increasingly effective countermeasures, it may become very difficult to perform many medical and surgical procedures that were challenging to perform prior to the commercialization of traditional antibiotics. In addition, secondary infections by MDR bacteria may increase mortality in conditions that are already difficult to manage such as COVID-19 and other diseases requiring hospitalization on a massive scale [3,4], which increase the risk for MDR-related infections [5,6,7]. Thus, there is an urgency to develop new antimicrobial strategies to address the pressing problems associated with MDR-related infectious diseases. Expanding the impact of antimicrobial resistance is the decrease in the pharmaceutical industry research pipeline for novel antimicrobial agents. In light of these challenges, several federal agencies and other organizations with specific programs to combat antibiotic resistance have inspired hope for novel antimicrobial discoveries. These organizations have formed partnerships with drug development companies to facilitate or accelerate pre-clinical and clinical testing requirements for novel antimicrobial agents. They include, but are not limited to, the Broad Spectrum Antimicrobials program of the Biomedical Advanced Research and Development Authority (BARDA), Combating Antibiotic Resistant Bacteria (CARB-X), European Gram Negative Antibacterial Engine (ENABLE), and the Repair Impact Fund. Noteworthy is the Joint Programming Initiative on Antimicrobial Resistance, which established the Virtual Research Institute (JPIAMR/VRI) with the mission to improve and coordinate research networks on antimicrobial resistance in seven countries in Europe and North America. Accordingly, the programs established by these organizations may create new funding opportunities for a robust exploration of antimicrobial peptides (AMPs), a promising source of new antimicrobial agents. It is in this context that we revisit some of the most promising data on engineered cationic antimicrobial peptides (eCAPs) to reframe efforts on the advances of helical AMPs toward clinical development.

## 2. Properties and Limitations of Natural Cationic Antimicrobial Peptides (AMPs)

To appreciate the logic in engineering synthetic AMPs, it is essential to understand the structure–function relationship of AMPs and the obstacles that have prevented their use in clinical treatment. Since their initial discoveries in the 1980s [8,9,10], AMPs have been shown to occur in almost all life forms as short peptides (10–50 amino acids long) with an amphipathic (e.g., cationic or anionic and hydrophobic domains) structure in the context of host defense [11,12,13,14]. Although classical AMPs are ribosomally synthesized (composed of the standard proteinogenic amino acids) [15,16], AMPs can be extended to include well-known peptides that display a cationic (or the less commonly known anionic) amphipathic structure (e.g., the cationic lipopeptide polymyxins or the anionic daptomycin) [17,18,19]. Hence, ubiquitous in nature, AMPs represent the first line of defense against a variety of microbial pathogens (e.g., bacteria, fungi, parasites, viruses), depending on amino acid composition of the amphipathic motif [20,21,22,23,24]. AMPs are structurally diverse (α-helix, β-sheets, loop structures, cyclic, etc.), indicating that the cationic amphipathic structure, rather than a particular secondary structure, is the most fundamental determinant of activity. In a classical sense, the most well-known structural classes are the α-helix and β-sheets through the pioneering work of Lehrer, Ganz, Boman, Zasloff, Hancock, and others [8,9,10,25,26,27]. In contrast, AMPs synthesized through extraribosomal pathways (e.g., the cyclic lipopeptides, polymyxins) [28,29], were known for several decades prior to the discovery of ribosomally synthesized AMPs [13,30,31,32]. While AMPs like the polymyxins and daptomycin are widely used clinically, those made exclusively of some of the 20 (excluding selenocysteine) conventional amino acids (including engineered derivatives of these AMPs) are yet to be clinically available. An important group comprises the primate theta-defensins (e.g., retrocyclins), not found in humans due to a premature stop codon [33,34]. In general, cyclic AMPs are significant for their enhanced stability compared to non-cyclic peptides, which partly explains a considerable interest in their clinical development [35,36,37,38]. Helical structures can be represented by magainins, cecropins, cathelicidins, and others [9,10,39,40,41]. β-sheet structures are exemplified by the α- and β-defensins (theta-defensins already mentioned), a highly ubiquitous class of AMPs with characteristic disulfide bridges that stabilize their secondary structures [26,42]. Importantly, there are many groups of AMPs that fall into one or more structural classes (including loop structures) and are well described elsewhere [43,44,45,46]. Cationic AMPs generally recognize their bacterial targets via electrostatic interactions with negatively charged bacterial membrane lipids [47,48,49,50,51,52,53,54,55,56]. This bacterial recognition is typically due to the cationic content in the context of the amphipathic structure and may result in the perturbation of the bacterial membrane in a concentration-dependent manner. Notably, the polymyxins are well known for their interactions with lipopolysaccharide (LPS), and that recognition may be reduced or abrogated with LPS modification (e.g., addition of ethanolamine or LPS deficiency) [57,58,59]. Daptomycin is another clinically used lipopeptide that kills gram-positive organisms (particularly *S. aureus*); it also permeabilizes bacterial membranes [60,61,62]. However, other antimicrobial mechanisms of AMPs have been demonstrated including bacterial cell penetration (e.g., proline-rich AMPs) and interference with vital intracellular processes [63,64,65,66]. Further, new antimicrobial mechanisms have been demonstrated in the last decade, notably the LPS-transport protein D (LptD) targeted by the cyclic peptide murepavadin [67,68,69,70].

AMPs are considered a highly promising therapeutic resource in the fight against bacterial resistance to traditional antibiotics because they have a number of properties that confer the ability to overcome the common resistance mechanisms of MDR pathogens. AMPs do not require metabolic processes for antimicrobial activity [71,72], as they target a pre-existing bacterial cell structure: the bacterial membrane. Therefore, they are effective against both quiescent and actively growing bacteria. As previously shown, the human AMP LL37 (Figure 1) rapidly kills *P. aeruginosa* in phosphate-buffered saline where there is no carbon source to support bacterial growth or an active metabolic state [55]. The majority of the bacterial cells were killed within the first minute, and killing of all 10^6^ cfu/mL was complete by 2.5 min. Thus, such rapid killing (less time for bacteria to divide and generate mutations) and disruption of a vital pre-existing structure (the bacterial membrane) make it difficult for mutations (requiring extensive bacterial division) to occur during AMP treatment and for ensuing selection of resistance compared to traditional antibiotics, which target specific metabolic pathways. Hence, in sharp contrast to conventional antibiotics, AMPs demonstrate a low propensity to select for bacterial resistance when bacterial cells grow in the presence of subinhibitory concentrations of AMPs [55,73,74,75,76,77]. A plausible explanation is that a fundamental reduction in electronegative density of the bacterial surface that would readily confer resistance to AMPs is likely to affect membrane integrity or fitness of the cells, thus rendering resistance to AMPs less common, although not impossible. Of note, resistance to several AMPs has been observed, including resistance against polymyxins [76,78,79], daptomycin [60,62,80,81], LL37 [73,82], and other AMPs [83,84,85].

AMPs have other interesting properties as well. The ability of select AMPs to display activity against multiple types of organisms (e.g., bacteria, viruses, fungi, and parasites) indicates the potential for application to a wide range of communicable or polymicrobial diseases [71,86,87,88,89,90,91,92,93,94]. In the context of the COVID-19 pandemic, the exploration of antiviral activity of AMPs is essential [91,95], as their antiviral mechanisms are not as well known as the electrostatic recognition of their target bacterial cell membranes. AMPs may also elicit an anti-infective host immune response and possess the ability to neutralize endotoxins, suggesting potential efficacy in septic shock [51,56,96,97,98,99,100]. Further, their anti-biofilm properties may confer efficacy against infections associated with wounds, medical implants, and chronic respiratory illnesses [82,91,101,102,103,104]. Antitumor properties also suggest a potential application against cancer [105]. Evidently, AMPs have the potential to display broad-spectrum efficacy in a wide range of applications, although some subsets of microorganisms may display various degrees of susceptibility to AMPs. An important trend in AMP development is the potential of AMPs to enhance the efficacy of traditional antibiotics when used in combination. Such synergistic properties may considerably contribute to the fight against antibiotic resistance by rendering the already-existing antibiotic arsenal more effective against MDR bacteria, as demonstrated by the polymyxins, murepavadin, and helical AMPs [106,107,108,109,110].

## 3. Overcoming the Limitations of AMPs

AMPs display several limitations that have delayed their successful development for clinical use.

### 3.1. Contextual Activity

Over the last several decades and since the early years of AMP discoveries, it has become increasingly clear that certain test conditions are challenging to most natural AMPs. Most classical AMPs typically demonstrate optimal bacterial killing activity in phosphate buffer, whereas their activity could be inhibited in the presence of varying concentrations of salt [55,73,75,77]. AMPs tend to show reduced activity under acidic conditions and in blood, plasma, or serum as they may bind to plasma proteins [55,75]. Therefore, in the context of a vast literature on antimicrobial properties in vitro, evidence for in vivo efficacy is still rare after almost four decades of discoveries. Furthermore, due to their peptidic property, AMPs are susceptible to protease digestion or even poor absorption (e.g., colistin in intestinal absorption) [111], which may limit their clinical applications only to parenteral routes of administration.

In addition to the contextual activity of many AMPs, there are other important concerns such as unclear pharmacokinetic (PK) properties, potential immunogenicity, and host toxicity. Protease digestion can be addressed with l-to-d enantiomerization and the use of unnatural amino acids or peptoids and other types of amino acid mimics [100,112,113,114,115,116,117]. Importantly, the short sequences of AMPs provide the advantage of conferring poor immunogenicity; however, it is important to rule it out for specific AMPs in clinical development. Another significant concern is the potential for host toxicity. Altogether, we and others have demonstrated that most of these limitations can be significantly overcome by structural design optimization, as shown by engineered peptides [28,38,55,75,104,118,119,120].

### 3.2. Advances in Helical AMP Engineering: eCAPs

There are, of course, several pioneers who paved the way for our development of engineered helical peptides by their outstanding work (e.g., Boman, Zasloff, and others), as previously mentioned. These early and subsequent discoveries as well as our studies of the last fifteen years have led to the understanding that the cationic amphipathic structure of helical AMPs is a consensus motif necessary for substantial activity [10,55,75,121,122]. In the late 1990s, the Mietzner–Montelaro group began a number of structure–function studies of the human immunodeficiency virus 1 (HIV-1) envelope proteins, specifically the C-terminal tail of HIV-1 transmembrane protein gp41 that contains two domains, termed lentivirus lytic peptides (LLPs), whose sequences have characteristics that are similar to those of natural AMPs (Figure 2). Secondary structure predictions suggested that the LLP sequences would form cationic amphipathic helices, an important characteristic of a large class of natural AMPs. These LLP analogs killed both gram-negative and gram-positive bacteria in vitro at low μM concentrations, confirming predicted antibacterial properties [41,123]. Further studies of these LLP structures revealed a selective incorporation and conservation of arginine, not lysine, suggesting a critically unique role for arginine in the LLP. Mutational studies of LLPs showed that membrane perturbation and antibacterial properties could be enhanced by adding Arg residues to the hydrophilic side of the helix, Trp residues to the hydrophobic side, and extending the length of the amphipathic helix via computationally optimized sequences using Arg, Val, and Trp [41,123,124,125,126]. These initial studies produced the first generation of *de novo*-engineered peptides from our group, which we referred to as eCAPs; hence, these studies initiated some of the earliest paradigms for engineering synthetic AMPs to improve activity against specific pathogens. Based on the LLP and parallel structure–function studies by the research groups of McLaughlin [127] and Degrado [128,129], we concluded that the remarkable diversity of natural AMPs reflects the fact that the evolution of specific structures of AMPs is highly influenced by host–pathogens interactions in the context of a particular biological environment, referred to as contextual activity of AMPs. An illustration of this restriction to the environment is the suppression of AMP activity observed in the lungs of cystic fibrosis (CF) patients associated with changes (e.g., changes in salt concentration and pH) in surface secretions [130,131]. Thus, when one either changes the target pathogen or environment (e.g., normal mucosal surface, abnormal airway, blood), AMP activity is diminished, as shown by the differential activities of LL37 against *P. aeruginosa* and *S. aureus* (Figure 3) in the presence or absence of saline.

An important concept is the generally accepted view that antibacterial activity is not due to a particular consensus sequence but rather to the secondary motif of a cationic amphipathic structure. The database of thousands of AMPs strongly supports the notion of consensus amphipathic secondary structure vs. consensus sequence [43,132,133]. However, prior to the discoveries of all these peptides, it was not that obvious. Thus, we initially addressed this hypothesis by reducing the diversity in amino acid composition observed in natural AMPs (e.g., 14 different amino acids in the human AMP LL37, Figure 1) to just two (LBU2, R and V; WR12, R and W) or three (WLBU2, R, W, and V) different amino acids (Figure 3). A diverse amino acid composition, while determining multiple functions displayed by LL37 (Figure 1) [134,135,136,137,138,139,140] is certainly not necessary to establish a cationic amphipathic motif. In fact, it may interfere with the idealization of a cationic amphipathic structure. Thus, LL37 structure includes 11 positively charged residues, but it has a net charge of +6 due to five electronegative residues. In addition, the hydrophilic and hydrophobic domains are not completely segregated compared to an idealized helical structure revealed by helical wheel analysis. In contrast, LBU2, a peptide made only of Arg and Val, requires a length of only 24 residues (compared to 37 in LL37) to display an antibacterial activity profile similar to that of LL37, including the reduced activity against *S. aureus* in phosphate-buffered saline (PBS) compared to activity in phosphate buffer alone [75]. Importantly, our group and others have shown that such a susceptibility to salt could be overcome by sequence optimization within the framework of cationic amphipathicity. Hence, in sharp contrast to LL37 and LBU2, Trp-rich eCAPs WLBU2 (made of Arg, Trp, and Val) and WR12 (made of Arg and Trp) retain activity in saline, divalent cations, human blood, and even acidic pH [55,71,75,77]. Of note, the eCAPs kill bacteria by rapidly disrupting the membrane as shown by biochemical assays (access of the impermeant ortho-Nitrophenyl-β-galactoside (ONPG) to the intracellular enzyme β-galactosidase in *P. aeruginosa*) [71] and transmission electron microscopy as well as propidium iodide (PI) incorporation of peptide-treated bacteria (unpublished data).

### 3.3. Overcoming Resistance by Recalcitrant Bacteria

A critical consideration in overall antimicrobial effectiveness is the potential for bacteria to evolve resistance to the agent(s) of interest, regardless of prior exposure to the antibiotic [18]. The molecular target of a traditional antibiotic requires a vital cellular process. Thus, for such an antibiotic to be effective, the bacterial cell must be in an active metabolic state that is advancing the life cycle of the cell. Ironically, an active metabolic state may permit the formation of new resistance-conferring mutations in the cell (during DNA replication). By contrast, a membrane-active antibiotic (e.g, WLBU2) targets a pre-existing cellular structure (e.g., LPS) and its activity is not necessarily affected by the metabolic state of the cell (no carbon source needed to be active). Similarly, the rapid membrane perturbation mechanisms of many helical AMPs are based on interactions with membrane lipids that may be essential to bacterial survival [71,77]. Resistance to AMPs typically occur by lipid modification, which reduces the net negative charge on the bacterial cell surface, while other resistance mechanisms are relatively less common (e.g., peptide digestion, inactivation, efflux) [141,142,143,144]. This membrane-active mechanism is the reason colistin is effective against MDR gram-negative bacteria (GNB) as a drug of last resort. Notably, this property is highly enhanced in rationally designed helical peptides [119,120,145,146]. The eCAPs WLBU2 and WR12 demonstrate effective antimicrobial activity against diverse clinical isolates of the ESKAPE pathogens (*Enterococcus faecium*, *Staphylococcus aureus*, *Klebsiella pneumoniae (K. pneumoniae)*, *Acinetobacter baumannii*, *Pseudomonas aeruginosa (P. aeruginosa)*, and *Enterobacter* species) that are resistant to LL37 and colistin and display lower propensity to select for resistant bacteria in vitro [2,73,74,147,148]. When bacteria are allowed to grow in the absence of selective pressure, no test strains resistant to Trp-rich eCAPs have been found (unpublished data). Once bacteria are allowed to grow by serial daily passages in the presence of sub-inhibitory concentrations of antibiotics or eCAPs, resistance to WLBU2 or WR12 was delayed by 28 days (resistance achieved within 28 days) in comparison to colistin (13 days) and traditional antibiotics (two days). These data are consistent with several reports on propensity of engineered AMPs to select for resistant bacteria compared to traditional antibiotics [73,120]. In addition, resistance to LL37 and colistin by ESKAPE pathogens typically does not result in resistance to the eCAPs. These results indicate the ability of these eCAPs to overcome the most common resistance mechanisms against current antibiotics and natural AMPs. However, while very uncommon, cross-resistance among these three types of membrane-active antimicrobial agents has been observed in our studies. In addition, although the differential mechanisms of AMPs and other antibiotics targeting specific metabolic pathways are obvious, the ability to remain active against colistin- and LL37-resistant strains does not necessary indicate a fundamentally different mechanism. Similarly, two drugs displaying different affinities for the same target does not necessarily mean their molecular targets are different. Differences in potency are not necessarily due to differences in mechanisms of action. In the case of optimized eCAPs, it could be that differences in potency are due to differential sequence optimization, using Trp in the context of the cationic amphipathic structure, resulting in enhanced effects on the bacterial membrane and cell death.

### 3.4. More Advanced Antimicrobial Properties

One of the biggest challenges to AMP development is their lack of in vivo efficacy, and this is consistent with their contextual activity in vitro. In that regard, eCAPs are advanced compared to most other AMPs. Similar to the development of pexiganan, most new AMPs in development continue to typically target topical applications. In contrast, more than a decade ago, we showed that systemic delivery was plausible. Aside from the in vivo efficacy in a vaginal *Chlamydia trachomatis* monkey model [149], WLBU2 demonstrates systemic efficacy in a murine *P. aeruginosa* septicemia model. This was the first evidence for systemic efficacy of a helical AMP [55,118]. To date, the idea of AMP development for systemic applications is still novel when compared to other advanced cationic helical AMPs in clinical trials for topical use (e.g., pexiganan, an analogue of magainin 2, which is a natural AMP isolated from frog skin) [150,151,152,153]. WLBU2 is able to protect mice against *P. aeruginosa* septicemia with a single dose of 3–4 mg/kg injected intravenously (i.v.) 1 h after or prior to *P. aeruginosa* exposure. Since WLBU2 displays a maximum tolerated dose (MTD) of 15–30 mg/kg in mice (slow i.v. injection) with a minimum therapeutic dose (mTd) of 3–4 mg/kg, the therapeutic index (TI = MTD/mTd) is ≤10). In that context, colistin, an antibiotic drug of last resort against infections due to MDR gram-negative bacteria (GNB), has a lower TI in similar animal models [154].

An important property of AMPs is the ability to prevent, disrupt, or eliminate bacterial biofilm growth. Surgical and medical device infections are typically associated with bacterial biofilms [155]. They are difficult to treat with traditional antibiotics due to the inherent tolerance of the biofilm to antibiotics. Traditional antibiotics require a high dose of antibiotics over an extended time period to eliminate biofilm [156,157]. As a result, antibiotics alone are often unable to treat an infection associated with a medical device, and removal of the implant is required [158,159,160]. WLBU2 displays activity against biofilm in both biotic and abiotic assays and against enveloped viruses either independently or in bacteria–virus co-culture models [82,91,101,102,104]. The anti-biofilm properties of eCAPs are further explored using *S. aureus* grown on a titanium rod and in murine models of trauma-associated infection. In comparison to cefazolin, WLBU2 was able to eradicate titanium-bound S. *aureus* antimicrobial resistant biofilm in vitro within 30 min [102]. This was the first evidence of a single antimicrobial agent capable of eliminating biofilm in a short time period. WLBU2 also displayed efficacy in an orthopedic implant biofilm animal model, indicating advanced antibiofilm properties relevant to saving implants and improving outcomes in these difficult-to-treat infections.

## 4. Perspective

Despite the advancement of eCAPs and other engineered AMPs toward clinical development [161], the work required to establish AMPs as novel antibiotics is still enormous. Even if WLBU2 becomes clinically available, some important concerns remain to be addressed. One is the increased risk of toxicity upon sequence optimization with the intent to increase antimicrobial potency. If the goal is to develop antibiotics that are sufficiently safe to use clinically, then the efficacy will be limited. Once a drug is being used for a relatively long period, bacteria will evolve to strains that are at least less susceptible. In the face of just a moderate decrease in susceptibility at best, the clinical use of such a drug will become severely limited, as the therapeutic window is not sufficiently wide to increase dosage in order to overcome such variations in susceptibility. The most effective way to overcome potential host toxicity of AMPs is to establish structure–function guidelines based on differential roles of the cationic and hydrophobic domains in the amphipathic structure of AMPs. Thus, it is important to determine how each domain uniquely contributes to microbial as opposed to host toxicity. Such structure–function studies may inform the prediction of amino acid composition, proportion of cationic and hydrophobic residues, length, and the arrangement of such residues that would more selectively target bacterial cells as opposed to host cells. Although some progress has been made in that regard, the studies have been incremental and not sufficiently systematic to provide guidelines that would substantially predict low risk of toxicity or other essential functions.

While the design of an AMP always begins with a rational basis, such design is typically implemented with a great deal of trial and error. To illustrate, let us compare WLBU2 with the most recent and promising AMPs. The structure of WLBU2 is based upon three main premises: (i) the optimization of the cationic amphipathic structure will enhance antibacterial potency; (ii) there must be an optimal length for highest antimicrobial potency; (iii) in the context of the helical amphipathic structure, Trp substitutions will enhance potency under physiological conditions [75]. Importantly, we did not have any information that would have predicted the sites for Trp substitution in relation to the potential toxicity to mammalian cells, except for the placement of the Trp residues in the hydrophobic domain of the secondary structure. Thus, while the design of WLBU2 began with a rational basis, an important part of the design resulted from an educated guess. The tendency for trial and error can also be demonstrated in AMP design that uses a template from a natural AMP for modification. As illustrated in Figure 4, two very recent AMPs were obtained from two natural amino acid sequences. Similarly to the use of Trp and few amino acids in WLBU2, SAAP-148 (LKRVWKRVFKLLKRYWRQLKKPVR), although derived from the cathelicidin LL37 (composed of 14 different amino acids), has an optimized cationic amphipathic structure with 24 residues in length [120]. This peptide also displays broad activity against MDR bacteria and moderate lytic effects on red blood cells at concentrations surpassing 10 µM. Similarly, the ZY4 peptide (VCKRWKKWKRKWKKWCV) was derived from a cathelicidin template with a very diverse amino acid composition, which was reduced to a composition of five amino acids (R, K, C, V, and W) and 17 residues in length. Interestingly, the cysteine residues provide a bridge that does not substantially disrupt the helical structure (an important lesson), as shown by the helical wheel projection (Figure 4) and by the three-dimensional (3D) structure [119]. In this case, the disulfide bridge may have a similar effect to that of stapling the peptide [162,163], which may contribute to its activity. Thus, both SAAP-148 and ZY4 show some structural similarities to WLBU2 and could not have been obtained without some level of trial and error 13–14 years after the first study on WLBU2 was published. These three examples (Figure 4) illustrate conceptually the incremental progress in the design of helical AMPs. Therefore, more transformative structure–function studies are necessary despite bold efforts in clinical development (LL37 for melanoma) [161]. Of note, the use of D-enantiomerization and unnatural amino acids, including peptoids, and cyclization of non-helical AMPs are additional strategies that could be effective in enhancing the pharmacological utility of AMPs [100,114,115,116].

## 5. Conclusions

While the field of AMPs is not progressing as fast as initially expected, the promising therapeutic impact on MDR-related infections is still relevant and the need for AMP development even more urgent. The eCAP and other AMP engineering technologies have not advanced the field fast enough amid lack of federal funding due to skepticism stemming from many limitations of AMPs, including past failed attempts at clinical development in previous decades. However, efforts to revive the field are still on the way, as several AMPs are in clinical trials [161], including WLBU2 (PLG0206), now in phase I trials for treatment of prosthetic joint infection associated with gram-positive bacteria. Similar to other AMPs in clinical trials [164], WLBU2 could become one of the first success stories carried out to completion. If so, the perception that classical AMPs are failed antimicrobial agents will change, which will open up more funding opportunities for consistent efforts to turn the prospect of AMPs as effective antimicrobials into a reality.

## 6. Patents

R.C.M. holds a patent on WLBU2 (PLG0206), which is licensed by Peptilogics.

## Figures and Tables

**Figure 1 pharmaceutics-12-00501-f001:**
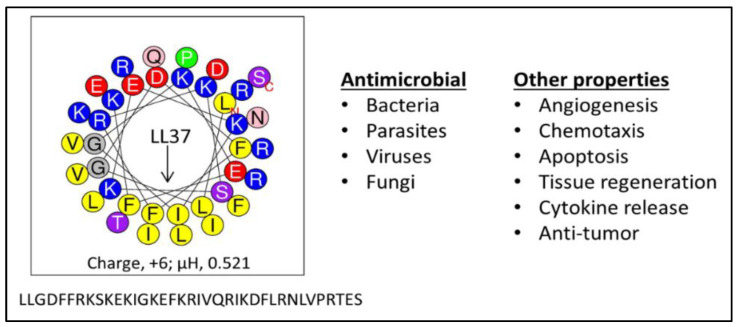
The human antimicrobial peptide (AMP) LL37. Composed of 14 different amino acids arranged to form a partial amphipathic helix (helical wheel) of 37 residues, LL37 (**left**) has evolved to display multiple functions (**right**); µH, hydrophobic moment, a measure of amphipathicity.

**Figure 2 pharmaceutics-12-00501-f002:**
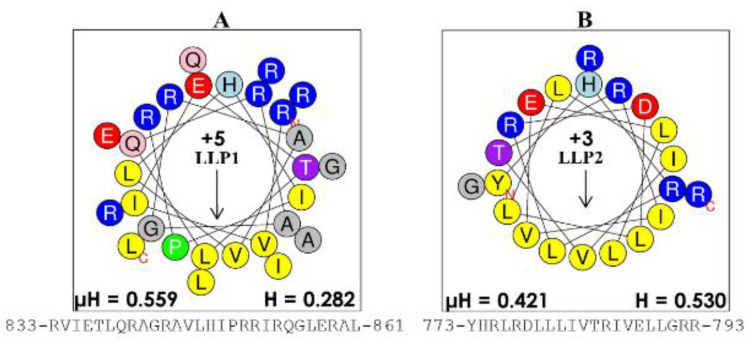
Structures of lentivirus lytic peptides (LLPs). The intracellular domains of the gp41 protein were identified as potential antimicrobial peptides because of their characteristic cationic amphipathic motif, with the helical wheel of (**A**) the more potent antibacterial LLP-1 (charge of +5) and (**B**) that of LLP-2 (charge of +3). These LLP domains set the foundation for the engineered AMPs, which we termed engineered cationic antimicrobial peptides (eCAPs) (Figure 3). The number on the left and the number on the right of each sequence indicate the beginning and the end of the primary sequence, respectively, in the protein gp41; µH, hydrophobic moment; H, hydrophobicity derived from the online software heliquest.fr.

**Figure 3 pharmaceutics-12-00501-f003:**
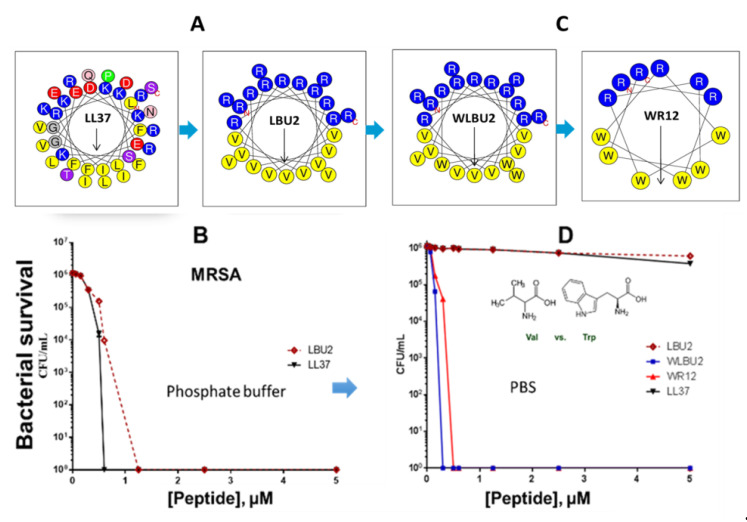
Progression of eCAP structures to overcome limitations of natural AMPs. The helical wheel of LL37 shows remarkable diversity in amino acid composition compared to eCAP LBU2 (**A**), which displays similar activity to LL37 (**B**). Sequence optimization of eCAPs (**C**) using Trp (W) resulted in resistance to saline (**D**) when the peptides were tested against methicillin-resistant *S. aureus* (MRSA, B and D). This figure is a summarized adaptation from previously published work [55,71,75,77].

**Figure 4 pharmaceutics-12-00501-f004:**
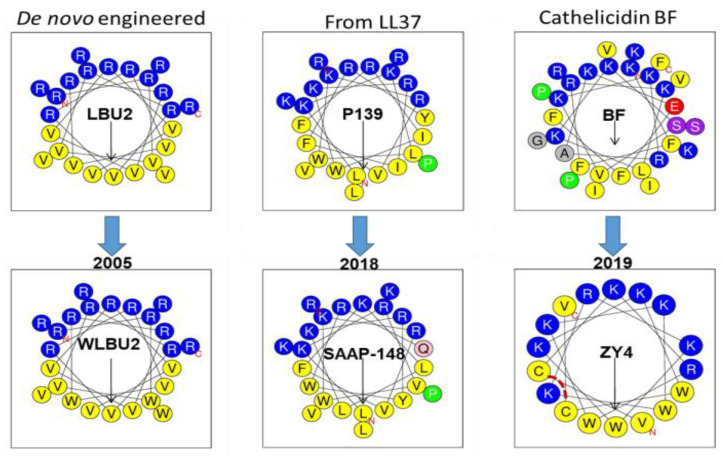
Comparison of the design of eCAP WLBU2 to that of recent AMPs with high potency. Both SAAP-148 and ZY4 are based on the modification of a cathelicidin with incorporation of Trp (W) and reduction in the number of amino acid residues to form a more ideal cationic amphipathic structure. ZY4 has a disulfide bridge indicated by a broken line in red.

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
