# Peer review of "Engineered Cationic Antimicrobial Peptides (eCAPs) to Combat Multidrug-Resistant Bacteria"

_pharmaceutics, 2020, doi:10.3390/pharmaceutics12060501_

Round 1

Reviewer 1 Report

The article by Deslouches et al. is a short overview mainly centered on the antimicrobial peptides developed by them (nicknamed eCAPs). Although not much is new, it might be a useful overview of this particular class of peptides. Hence, the paper may be suitable for publication in Pharmaceutics, but it needs to be improved and updated.

Comments:

Lines 40-42: Please provide a wider scope (or at least name them) to research and development programs in addition to BARDA: CARB-X, ENABLE (IMI 7FP), REPAIR Impact Fund (Novo Nordisk), JPI AMR.

Lines 52-55: The cyclic peptide section is very poorly described. In a few lines, cyclic peptides are dismissed and references do not seem the most adequate: they are either old or deal tangentially with the topic. Please, include (not exhaustive search) at least:

                - Nat. Prod. Res., 2017,34, 886; J. Med. Chem. 2010, 53, 1898,

Expert Opinion on biological therapy 17, 663-676, 2017;  Nature 576459–464 (2019), and others that authors may consider.

In my opinion, this section deserves more attention. As far as I know, cyclic antimicrobial peptides are the only peptide antibiotics available in the market, and the scientific field is really active (novel polymyxins and murepavidin in preclinical/clinical phases, for instance).

Line 80-82: Please add references regarding the well-studied mechanism of polymyxins and daptomycin resistance, i. e.:

Mechanisms of Drug Resistance: Daptomycin Resistance, Tran et al, Ann N Y Acad Sci. 2015 Sep; 1354: 32–53.

https://www.ncbi.nlm.nih.gov/pmc/articles/PMC4966536/

Polymyxins: Antibacterial Activity, Susceptibility Testing, and Resistance Mechanisms Encoded by Plasmids or Chromosome

Clinical Microbiology Reviews Mar 2017, 30 (2) 557-596; DOI: 10.1128/CMR.00064-16

 Line     108-110: “...limit their clinical applications only to parenteral routes of administration” add that AMPs also have a  poor oral absorption (if they are not proteolyzed, of course). Polymyxin B, for instance, is also used for selective digestive tract decontamination.

LIne 124, Please, include at least the pioneering work by Merrifield and Boman (both Nobel prizes) on helical amphipathic AMP cecropin, from 1982:

Synthesis of the antibacterial peptide cecropin A (1-33).

Merrifield RB, Vizioli LD, Boman HG. Biochemistry. 1982 Sep 28;21(20):5020-31.

https://pubs.acs.org/doi/abs/10.1021/bi00263a028

Line 301:”... cysteine residues provide a bridge that does not substantially disrupt the helical  structure (an important lesson), as shown by the helical wheel conformations and biophysical.”

The helical wheel projection (not conformation!) does not clearly illustrate such a statement to the reader. Authors should provide a 3D picture, for instance,  showing that the disulfide bond does not alter the helical structure (or a reference or CD spectrum). Which is the distance between both Cys residues? i, i+4 residues? The linear sequence of the peptide could also help.

Line 318: “WLBU2 is now in phase I trials for the treatment of knee infection”,  please give more details (Gram negative bacteria infection? Gram positive?).  Please, could you also provide a reference of the Clinical trial? I could not find it in the Pew Trusts list; is it under another name?

Author Response

Reviewer # 1

  1. Lines 40-42: Please provide a wider scope (or at least name them) to research and development programs in addition to BARDA: CARB-X, ENABLE (IMI 7FP), REPAIR Impact Fund (Novo Nordisk), JPI AMR.

We thank the reviewer for this insightful comment. We addressed it in Lines 39-53 of the revised manuscript:

In light of these challenges, several federal agencies and other organizations with specific programs to combat antibiotic resistance have inspired hope for novel antimicrobial discoveries. These organizations form partnerships with drug development companies to facilitate or accelerate pre-clinical and clinical testing requirements for novel antimicrobial agents. They include, but are not limited to, the Broad Spectrum Antimicrobials program of the Biomedical Advanced Research and Development Authority (BARDA), Combating Antibiotic Resistant Bacteria (CARB-X), European Gram Negative Antibacterial Engine (ENABLE), and Repair Impact Fund. Noteworthy is the Joint Programming Initiative on Antimicrobial Resistance, which established the Virtual Research Institute (JPIAMR/VRI) with the mission to improve and coordinate research networks on antimicrobial resistance in seven countries in Europe and North America. Accordingly, the programs established by these organizations may create new funding opportunities for a robust exploration of antimicrobial peptides (AMPs), a promising source of new antimicrobial agents. It is in this context that we revisited some of the most promising data on engineered cationic antimicrobial peptides (eCAPs) to reframe the efforts on the advances of helical AMPs toward clinical development.

  1. Lines 52-55: The cyclic peptide section is very poorly described. In a few lines, cyclic peptides are dismissed and references do not seem the most adequate: they are either old or deal tangentially with the topic. Please, include (not exhaustive search) at least:

                - Nat. Prod. Res., 2017,34, 886; J. Med. Chem. 2010, 53, 1898,

Expert Opinion on biological therapy 17, 663-676, 2017;  Nature 576, 459–464 (2019), and others that authors may consider.

In my opinion, this section deserves more attention. As far as I know, cyclic antimicrobial peptides are the only peptide antibiotics available in the market, and the scientific field is really active (novel polymyxins and murepavadin in preclinical/clinical phases, for instance).

We thank the reviewer for this comment and for the references, which are all included among other references in the following text (lines 65-82):

AMPs are structurally diverse (α-helix, β-sheets, loop structures, cyclic, etc.), indicating that the cationic amphipathic structure, rather than a particular secondary structure, is the most fundamental determinant of activity. In a classical sense, the most well-known structural classes are the α-helix and β-sheets through the pioneering work of Lehrer, Ganz, Boman, Zasloff, Hancock, and others [8-10,25,26]. In contrast, the non-classical AMPs (e.g., the cyclic lipopeptides, polymyxins) [27,28], were known for several decades prior to the discovery of ribosomally synthesized AMPs [13,29-31]. While AMPs like the polymyxins and daptomycin are widely used clinically, AMPs made exclusively of some of the 20 (excluding selenocysteine) conventional amino acids (including those modeled after these AMPs) are yet to be clinically available. Among them are the primate theta-defensins (e.g., retrocyclins, not found in human due to a premature stop codon) [32,33]. In general, cyclic AMPs are significant for their enhanced stability compared to the non-cyclic peptides, which partly explains a considerable interest in their clinical development [34-37]. Helical structures can be represented by the magainins, cecropins, cathelicidins, and others [9,10,38-40]. β-sheet structures are exemplified by the α- and β-defensins (theta-defensins already mentioned), a highly ubiquitous class of AMPs with characteristic disulfide bridges that stabilize their secondary structures [26,41]. Importantly, there are many groups of AMPs that fall into one or more structural classes (including loop structures) and are well described elsewhere [42-45].

  1. Line 80-82: Please add references regarding the well-studied mechanism of polymyxins and daptomycin resistance, i. e.:

Mechanisms of Drug Resistance: Daptomycin Resistance, Tran et al, Ann N Y Acad Sci. 2015 Sep; 1354: 32–53.

https://www.ncbi.nlm.nih.gov/pmc/articles/PMC4966536/

Polymyxins: Antibacterial Activity, Susceptibility Testing, and Resistance Mechanisms Encoded by Plasmids or Chromosome

Clinical Microbiology Reviews Mar 2017, 30 (2) 557-596; DOI: 10.1128/CMR.00064-16

We appreciate these references and they are included in the text (ref 61 and 76).

  1. Line 108-110: “...limit their clinical applications only to parenteral routes of administration” add that AMPs also have a  poor oral absorption (if they are not proteolyzed, of course). Polymyxin B, for instance, is also used for selective digestive tract decontamination.

We thank the reviewer for this comment, which is addressed in lines 143-146:

 Furthermore, due to their peptidic property, AMPs are susceptible to protease digestion or even poor absorption (e.g., colistin intestinal absorption) [107], which may limit their clinical applications only to parenteral routes of administration.

  1. LIne 124, Please, include at least the pioneering work by Merrifield and Boman (both Nobel prizes) on helical amphipathic AMP cecropin, from 1982:

Synthesis of the antibacterial peptide cecropin A (1-33).

Merrifield RB, Vizioli LD, Boman HG. Biochemistry. 1982 Sep 28;21(20):5020-31.

https://pubs.acs.org/doi/abs/10.1021/bi00263a028

We thank the reviewer for this reference, which is included as ref # 10.

  1. Line 301:”... cysteine residues provide a bridge that does not substantially disrupt the helical structure (an important lesson), as shown by the helical wheel conformations and biophysical.”

The helical wheel projection (not conformation!) does not clearly illustrate such a statement to the reader. Authors should provide a 3D picture, for instance, showing that the disulfide bond does not alter the helical structure (or a reference or CD spectrum). Which is the distance between both Cys residues? i, i+4 residues? The linear sequence of the peptide could also help.

We thank the reviewer for this comment. The 3D structure of the ZY4 peptide is published (Ref 158, Figure 1B). We included the sequence in the text as follows:

Similarly, the ZY4 peptide (VCKRWKKWKRKWKKWCV) was derived from a cathelicidin template with a very diverse amino acid composition, which was reduced to a composition of five amino acids (R, K, C, V, and W) and 17 residues in length. Interestingly, the cysteine residues provide a bridge that does not substantially disrupt the helical structure (an important lesson), as shown by the helical wheel projection (Figure 4) and supported by the 3D structure [158].

  1. Line 318: “WLBU2 is now in phase I trials for the treatment of knee infection”, please give more details (Gram negative bacteria infection? Gram positive?).  Please, could you also provide a reference of the Clinical trial? I could not find it in the Pew Trusts list; is it under another name?

This is a very insightful of the reviewer. Addressing this comment brings more clarity to this matter as described in lines 340-346:

However, efforts to revive the field are still on the way, as several AMPs are in clinical trials [157] including WLBU2 (PLG0206), now in phase I trials for treatment of prosthetic joint infection associated with gram-positive bacteria. Similar to many other AMPs in clinical trials [161], WLBU2 could become one of the first success stories carried out to completion. If so, the perception that AMPs are failed antimicrobial agents will change, which will open up more funding opportunities for consistent efforts to turn the prospect of AMPs as effective antimicrobials into a reality.

Reviewer 2 Report

This is a well written review that will be of interest to the readership of your journal.  I have the following suggestions that would strengthen this level of interest.

  1. While, I applaud the authors for their valuable contributions to this field, my personal feeling is that this manuscript should simply be a review of this topic, without the implication that the primary goal of the review is to review the authors’ work. Some of the descriptions used “we reviewed our data of the last 15 years…” seems to somewhat trivialize the overall breadth of literature on this topic.
  2. A brief description of defensins and perhaps retrocyclins in the introduction would be of value.
  3. A description of using AMPs in combination with antibiotics and other AMPs should also be added, as this approach seems to be gaining traction and will likely lead to success in vivo. See below for a few recent examples of this concept.

    Biochim Biophys Acta Biomembr. 2019 Oct 1;1861(10):182980. doi: 10.1016/j.bbamem.2019.05.002. Epub 2019 May 5. Hybrids made from antimicrobial peptides with different mechanisms of action show enhanced membrane permeabilization.Wade HM1, Darling LEO2, Elmore DE3.

    Antibiotics (Basel). 2020 Mar 20;9(3). pii: E128. doi: 10.3390/antibiotics9030128. Pexiganan in Combination with Nisin to Control Polymicrobial Diabetic Foot Infections.

    Gomes D1, Santos R1, S Soares R1, Reis S1, Carvalho S1, Rego P2, C Peleteiro M1, Tavares L1, Oliveira M1.

    Sci Rep. 2020 Feb 27;10(1):3580. doi: 10.1038/s41598-020-60570-w.Plantaricin NC8 αβ exerts potent antimicrobial activity against Staphylococcus spp. and enhances the effects of antibiotics. Bengtsson T1, Selegård R1,2, Musa A1, Hultenby K3, Utterström J2, Sivlér P4, Skog M4, Nayeri F5, Hellmark B6, Söderquist B1,6, Aili D2, Khalaf H7.

    Microb Path Combinations of early generation antibiotics and antimicrobial peptides are effective against a broad spectrum of bacterial biothreat agents.Cote CK, Blanco II, Hunter M, Shoe JL, Klimko CP, Panchal RG, Welkos SL.Microb Pathog. 2020 Feb 9;142:104050. doi: 10.1016/j.micpath.2020.104050. [Epub ahead of print]

MINOR POINTS

Line 236. Change to “Chlamydia trachomatis

Line 270. Avoid contractions “isn’t” should be “is not”

Line 284. Should read “WLBU2 is based upon the…”

Author Response

Reviewer # 2

  1. While, I applaud the authors for their valuable contributions to this field, my personal feeling is that this manuscript should simply be a review of this topic, without the implication that the primary goal of the review is to review the authors’ work. Some of the descriptions used “we reviewed our data of the last 15 years…” seems to somewhat trivialize the overall breadth of literature on this topic.

We appreciate the thoughtfulness of the reviewer, and we revised the language in Line 51-53:

It is in this context that we revisited some of the most promising data on engineered cationic antimicrobial peptides (eCAPs) to reframe the efforts on the advances of helical AMPs toward clinical development.

  1. A brief description of defensins and perhaps retrocyclins in the introduction would be of value.

We thank the reviewer for this comment, which is addressed in Lines 67-80:

In a classical sense, the most well-known structural classes are the α-helix and β-sheets through the pioneering work of Lehrer, Ganz, Boman, Zasloff, and others [8-10,25,26]. In contrast, AMPs synthesized through extraribosomal pathways (e.g., the cyclic lipopeptides, polymyxins) [27,28], were known for several decades prior to the discovery of ribosomally synthesized AMPs [13,29-31]. While AMPs like the polymyxins and daptomycin are widely used clinically, those made exclusively of some of the 20 (excluding selenocysteine) conventional amino acids (including engineered derivatives of these AMPs) are yet to be clinically available. An important group comprises the primate theta-defensins (e.g., retrocyclins), not found in human due to a premature stop codon) [32,33]. In general, cyclic AMPs are significant for their enhanced stability compared to the non-cyclic peptides, which partly explains a considerable interest in their clinical development [34-37]. Helical structures can be represented by the magainins, cecropins, cathelicidins, and others [9,10,38-40]. β-sheet structures are exemplified by the α- and β-defensins (theta-defensins already mentioned), a highly ubiquitous class of AMPs with characteristic disulfide bridges that stabilize their secondary structures [26,41].

  1. “A description of using AMPs in combination with antibiotics and other AMPs should also be added, as this approach seems to be gaining traction and will likely lead to success in vivo. See below for a few recent examples of this concept.

Biochim Biophys Acta Biomembr. 2019 Oct 1;1861(10):182980. doi: 10.1016/j.bbamem.2019.05.002. Epub 2019 May 5. Hybrids made from antimicrobial peptides with different mechanisms of action show enhanced membrane permeabilization.Wade HM1, Darling LEO2, Elmore DE3.

Antibiotics (Basel). 2020 Mar 20;9(3). pii: E128. doi: 10.3390/antibiotics9030128. Pexiganan in Combination with Nisin to Control Polymicrobial Diabetic Foot Infections.

Gomes D1, Santos R1, S Soares R1, Reis S1, Carvalho S1, Rego P2, C Peleteiro M1, Tavares L1, Oliveira M1.

Sci Rep. 2020 Feb 27;10(1):3580. doi: 10.1038/s41598-020-60570-w.Plantaricin NC8 αβ exerts potent antimicrobial activity against Staphylococcus spp. and enhances the effects of antibiotics. Bengtsson T1, Selegård R1,2, Musa A1, Hultenby K3, Utterström J2, Sivlér P4, Skog M4, Nayeri F5, Hellmark B6, Söderquist B1,6, Aili D2, Khalaf H7.

Microb Path Combinations of early generation antibiotics and antimicrobial peptides are effective against a broad spectrum of bacterial biothreat agents.Cote CK, Blanco II, Hunter M, Shoe JL, Klimko CP, Panchal RG, Welkos SL.Microb Pathog. 2020 Feb 9;142:104050. doi: 10.1016/j.micpath.2020.104050. [Epub ahead of print]”

We appreciate the reviewer’s insight and included all these references, as shown in lines 125-130:

An important trend in AMP development is the potential of AMPs to enhance the efficacy of traditional antibiotics when used in combination. Such synergistic properties may considerably contribute to the fight against antibiotic resistance by rendering the already-existing antibiotic arsenal more effective against MDR bacteria, as demonstrated by the polymyxins, Murepavadin, and helical AMPs [104-108].

  1. MINOR POINTS

Line 236. Change to “Chlamydia trachomatis”

Line 258: Aside from the in vivo efficacy in a vaginal Chlamydia trachomatis monkey model [93], WLBU2 demonstrates systemic efficacy in a murine P. aeruginosa septicemia model. This was the first evidence for systemic efficacy of an AMP [31,70]

Line 270. Avoid contractions “isn’t” should be “is not”

Lines 292-294: In the face of just a moderate decrease in susceptibility at best, the clinical use of such a drug will become severely limited, as the therapeutic window is not sufficiently wide to increase dosage in order to overcome such variations in susceptibility

Line 284. Should read “WLBU2 is based upon the…”

Line 306: The structure of WLBU2 is based upon the three main premises.

Reviewer 3 Report

The manuscript is explained in a logical format. The authors start with talking about Engineered cationic peptides (eCAPs) with examples. WLBU2 has been explained through the projection of the helical wheel, which was easy to understand, and provided an example with the modification with Trp, which increased the activity. The authors have given a logical perspective on how to improve the design and synthesis of AMPs and how to overcome the toxicity to molecules in the clinic. I can recommend the manuscript for publication with minor corrections. The following suggestions could be used to improve the manuscript to increase its appeal to wider readership of the journal.

Minor corrections

The manuscript lacks clarity on basic concepts. The brief explanations of these concepts as a visual diagram, Such as Mechanism of action of common AMPs, Structures of AMPs could be useful for readers. The authors should check for the consistency throughout the manuscript for example gram positive is usually presented as Gram positive.

It would be useful to add clarification on why eCAP develop slow resistance than polymyxin or other cationic peptides?

Please add the reference for implant biofilm animal model, to published work in figure 3.

Line 53 – The Authors claims that classical AMPs are ribosomally synthesised. Is this statement true, as a good understanding that these are also synthesised by NRPS (non-ribosome peptide synthetase) and PKS (polyketide synthetase)? Would you please add reference/s for this?

Figure 1-Please indicate clearly (key legend) what the colour coding stand for e.g. (the blue are cationic amino acids)

Line 224 – “differences in potency are not necessarily due to differences in mechanisms of action”. This is of course a true statement, however this and the statement below “thus the modification of an AMP may enhance activity without necessarily changing the target” is very broad and general and should be avoided, unless if the authors want to give a precise detail with references.

Line 286 – the authors mention Trp substitutions will enhance potency under physiological conditions. Yes, this has been proven with WLBU2. Could authors provide any references that only Trp residues enhance activity as there are broad ranges of hydrophobic amino acid present, including non-proteinogenic Amino acid.

Line 193; This bit needs to be re-written. Antibiotics allow selection. Resistance do not occur mainly due to mutations as a result of exposure. Also, you need references here.

Line 199; Not particularly true (re polymyxin). There is mixing in the concepts here.

Line 209; do you mean resistant bacteria?

Line 210; re (using population-based selection of resistance): what type of surveillance did you carry? what was the case definition?

Line 213; re (markedly delayed): Does that mean they have showed tolerance at the end? what bacteria was tested? how delayed? how many days was this experiment and at which point they started to grow? describe the growth rate.

Line 215; re (AMPs to invoke selection of resistance compared to current antibiotics): This is mis-conceptualized. antibiotics do not invoke. They are selective pressures.

Line 219; re (uncommon, cross-resistance): So why this resistance would not be selected and mobilised?

Line 228; re (concentration-dependent): misleading. all antibiotic doses are concentration dependant.

Figure 3; Where is Figure 3 C and D description?

Also, format/ fonts are not consistent.

Author Response

Reviewer # 3

  1. The manuscript lacks clarity on basic concepts. The brief explanations of these concepts as a visual diagram, Such as Mechanism of action of common AMPs, Structures of AMPs could be useful for readers. The authors should check for the consistency throughout the manuscript for example gram positive is usually presented as Gram positive.

We appreciate the reviewer’s concern. After 40 years of discovery on AMPs, we think that there are many reviews that include such diagrams, and references are added to address the lack of such diagrams in the manuscript.

Regarding “gram positive” or gram negative, although it is actually common to erroneously capitalize “G” in published manuscripts, we prefer to follow the NIH guidelines (https://wwwnc.cdc.gov/eid/page/preferred-usage) and use gram positive/negative as opposed to “Gram” throughout the manuscript. We would have converted any Gram to gram, had we found such inconsistency in the manuscript. Gram begins with a “G” in “European Gram Negative Antibacterial Engine (ENABLE)” because it is part of the name of an organization.

  1. It would be useful to add clarification on why eCAP develop slow resistance than polymyxin or other cationic peptides?

Unfortunately, we do not know for sure, and this is an area of great research interest. We think it is because the eCAPs are optimized with Trp. We plan to test whether optimization with Trp as opposed to, for example, Leu or Phe would lead to different outcomes.

  1. Please add the reference for implant biofilm animal model, to published work in figure 3.

We think the reviewer for this comment. We added the references (54, 75, and 109) as suggested.

  1. Line 53 – The Authors claims that classical AMPs are ribosomally synthesised. Is this statement true, as a good understanding that these are also synthesised by NRPS (non-ribosome peptide synthetase) and PKS (polyketide synthetase)? Would you please add reference/s for this?

We appreciate the reviewer’s comment, as this distinction is only conceptual. What is only factual and also acknowledged by the reviewer is that some AMPs require ribosomal synthesis, whereas others do not. The reviewer can also agree that the terms antimicrobial peptides were initially dedicated to AMPs that are ribosomally synthesized with proteinogenic amino acids despite that AMPs like polymyxins were known for  several decades before realizing they had similar properties to the ribosomally synthesized AMPs discovered by Lehrer, Boman, Zasloff and others. From that perspective, I use the terms “classical AMPs” for clarity because there is a difference in structure, limitations, and clinical status.  To address the reviewer’s concern, we revised the text in lines 65-74:

AMPs are structurally diverse (α-helix, β-sheets, loop structures, cyclic, etc.), indicating that the cationic amphipathic structure, rather than a particular secondary structure, is the most fundamental determinant of activity. In a classical sense, the most well-known structural classes are the α-helix and β-sheets through the pioneering work of Lehrer, Ganz, Boman, Zasloff, and others [8-10,25,26]. In contrast, AMPs synthesized through extraribosomal pathways (e.g., the cyclic lipopeptides, polymyxins) [27,28], were known for several decades prior to the discovery of ribosomally synthesized AMPs [13,29-31]. While AMPs like the polymyxins and daptomycin are widely used clinically, those made exclusively of some of the 20 (excluding selenocysteine) conventional amino acids (including engineered derivatives of these AMPs) are yet to be clinically available.

  1. Line 224 – “differences in potency are not necessarily due to differences in mechanisms of action”. This is of course a true statement, however this and the statement below “thus the modification of an AMP may enhance activity without necessarily changing the target” is very broad and general and should be avoided, unless if the authors want to give a precise detail with references.

To address the reviewer’s comment, the following sentence is omitted in the revised manuscript: “Thus, modification of an AMP may enhance activity without necessarily changing the molecular target.”

  1. Line 286 – the authors mention Trp substitutions will enhance potency under physiological conditions. Yes, this has been proven with WLBU2. Could authors provide any references that only Trp residues enhance activity as there are broad ranges of hydrophobic amino acid present, including non-proteinogenic Amino acid.

We appreciate the reviewer’s comment and regret that we were not sufficiently clear about the impact of trp. We would never insinuate that only Trp would enhance activity. This is why we added in lines 308-309: “In the context of the helical amphipathic structure, Trp substitutions will enhance potency under physiological conditions.”

In future studies, we also plan to systematically replace Trp with other hydrophobic amino acids using pre-existing Trp-rich peptides. There are many unconventional amino acids that need to be considered, including other aromatic groups. In addition, there is no evidence that Trp would enhance activity across all structural classes of AMPs. These are some of the reasons why there is a lot to be done to further advance the field in that regard.

  1. Line 193; This bit needs to be re-written. Antibiotics allow selection. Resistance do not occur mainly due to mutations as a result of exposure. Also, you need references here.

We understand the reviewer’s concern. We refer to selection (not occurrence) of resistance during selective pressure as a matter of fact. We know that there are adaptations and growth conditions that increase antibiotic tolerance, and mutation is a result of DNA replication in the presence or absence of antibiotics. 

The original (lines 201-206) sentence states: “A critical consideration in overall antimicrobial effectiveness is the potential for bacteria to evolve resistance to the agent(s) of interest.”

We modified the sentence as follows (214-215:

“A critical consideration in overall antimicrobial effectiveness is the potential for bacteria to evolve resistance to the agent(s) of interest, regardless of prior exposure to the antibiotic.”

  1. Line 199; Not particularly true (re polymyxin). There is mixing in the concepts here.

It is not clear what is not true about polymyxins in these lines 187-200. However, we replaced polymyxins with WLBU2, guessing that the reviewer may object to the point that the metabolic state of the cell may not affect the activity of the polymyxins.

Revised in lines 219-223:

By contrast, a membrane-active antibiotic (e.g, WLBU2), targets a pre-existing cellular structure (e.g., LPS), and its activity is not necessarily affected by the metabolic state of the cell during exposure to such antibiotic (no carbon source needed to be active). Similarly, the rapid membrane perturbation mechanisms of many helical AMPs are based on interactions with membrane lipids that are essential to bacterial survival [37,41].

  1. Line 209; do you mean resistant bacteria?

Line 232: We changed bacterial resistance phenotype to “resistant bacteria”

  1. Line 210; re (using population-based selection of resistance): what type of surveillance did you carry? what was the case definition?

Here we were simply making the point when bacteria were growing in the absence of antimicrobial agent (no pressure to select for a particular strain among the growing population).  In case the term is confusing, we modified the sentence as follows (lines 232-233):

When bacteria are allowed to grow in the absence of selective pressure, no test strains resistant to Trp-rich eCAPs have been found (unpublished data).

  1. Line 213; re (markedly delayed): Does that mean they have showed tolerance at the end? what bacteria was tested? how delayed? how many days was this experiment and at which point they started to grow? describe the growth rate.

We changed the sentence as follows (lines 234-238): “When bacteria are allowed to grow by serial daily passages in the presence of sub-inhibitory concentration of antibiotics and eCAPs, resistance to WLBU2 or WR12 was delayed by 2 to 4 weeks in comparison to colistin and traditional antibiotics, respectively. Please refer to the reference # 72 for the detailed method.

  1. Line 228; re (concentration-dependent): misleading. all antibiotic doses are concentration dependant.

“…resulting in enhanced and distinct concentration-dependent effects on the bacterial membrane and cell death.”

Here we referred to the effects on bacterial membrane (not the doses) as being concentration-dependent. To address the concern of the reviewer, we removed “concentration-dependent” from the sentence as follows (lines 238-239): “…resulting in enhanced and distinct effects on the bacterial membrane and cell death.”

  1. Figure 3; Where is Figure 3 C and D description?

We are very thankful that the reviewer caught this error. We revised the figure legend as follows:

Figure 3. Progression of eCAP structures to overcome limitations of natural AMPs. The helical wheel of LL37 shows remarkable diversity in amino acid composition compared to eCAP LBU2 (A), which displays similar activity to LL37 (B). Sequence optimization of eCAPs (C) using Trp (W) resulted in resistance to saline (D) when the peptides were tested against S. aureus; this figure is a summarized adaptation from previously published work {55, 76, 111].